# Looking to Autoethnography as Spiritual Practice

**Wendy Anne Bilgen**

Independent Researcher, Cleveland, OH 44115, USA; wendybilgen@gmail.com

**Abstract:** In this autoethnographic account, I explore through creative analytic practice (CAP) how autoethnography has become a practice for spiritual connection. Inspired by Martin Buber's *I-Thou* philosophy, I discuss how autoethnographic encounters have the potential to emulate what is characterized by Buber as encounters with an *Eternal Thou*. In Buber's conceptualization, when an individual *I* encounters a *Thou*, in a dialogical relationship of mutual honor and embrace of shared humanity, an Eternal Thou encounter is experienced. The qualities of the I-Thou encounter are mirrored in autoethnography as the autoethnographer (*I*) and the reader/participant (*Thou*) share a numinous, sacred, spiritual exchange. Processes inherent to autoethnographic work enable the conditions wherein the sacredness of *I-Thou* encounters are nourished. I invite the reader to wonder with me about the ways that authentic, vulnerable, and critical engagement with one's own story paves the way for *I-Thou* experiences to be ignited and imaginative possibilities multiplied so that individual and social transformation might follow. Finally, I question whether autoethnography might help fill a gap in our current spiritual research paradigms.

**Keywords:** *I-Thou* connection; autoethnography; spiritual connection



## 1. Autoethnographic Method and Spiritual Explorations

Autoethnography is a research and writing practice in which a researcher examines a phenomenon in their own culture through their own experiences as one fully immersed in the cultural context of interest (Chang 2008). It is a self-reflexive, dialogical, and embodied exploration of body, mind, soul, and spirit with the self as participant-observer, positioned as both researcher and researched (Jones et al. 2013; Rafi 2021). The result is research and writing that creates "critical spaces" for dialogical and inter-relational exchanges to be made "between the storyteller/storymaker, her lived and embodied experiences, and the readers" (Metta 2016, p. 494). The product of an autoethnography is often creative, artistic, evocative, and transformative (Hernandez et al. 2022) inviting new insights and understandings of what it means to be human in the particular context under surveillance. That is quite a lot to extract from a process of inquiry. And now, with this article I suggest that autoethnography is all of that, and more; it can also be a practice that enhances spiritual connections.

I first began to wonder about the spiritual impact of the autoethnographic process while working on my autoethnographic dissertation. My study explored the formation of my leadership identity within the contexts of my spiritual and religious life raised in the culture of reformed Jewishness, embracing Christian teaching, and later living as a social worker/therapist with my Turkish husband in predominately Islamic Turkey (Bilgen 2018). The more I was immersed in the world of autoethnography, reading it and writing it, the more I began to notice the way that autoethnography itself had the capacity to powerfully illuminate, transform, and connect those who practice it. I also noticed that the words I was using to describe the autoethnographic experience were words that I previously had associated with spiritual encounters.

Through engaging the world autoethnographically, it began to seem like so much more than a method of inquiry, or as Bochner (2013) called it, a way of life. It had clearly taken on some of the characteristics of a spiritual practice for me. And so I questioned more intentionally, *what makes autoethnography so much like a spiritual connection experience? And*

*what might the benefit be of embracing it as such?* As autoethnographers do, I draw from my own experiences. I begin as a spiritually minded autoethnographer, exploring what I mean by saying the experience was "spiritual."

What I Mean by "Spiritual":

I embrace the reality that there is no singular way of describing or decisively defining what is or is not a spiritual experience[1] (Lin et al. 2016). If there was one way of defining it, there would be no question about where to look for spiritual experiences or what we are looking for. Instead, some may "connect with Spirit, in the arms of nature" (Poulos 2010, p. 51), while others find their craving for transcendence "satisfied by group life, groups ever widening, ever unifying, but always groups" (Quatro 2004, p. 229). Still, others may attempt to understand the spiritual through their senses, empirical evidence, or intuition, while others through religious ritual, or an acquisition of knowledge that affirms what and who God is (aka theology).

Undoubtedly, our religious upbringing and the religious contexts in our lives will also affect our views, beliefs, awareness, and openness to spiritual experiences (Tippett 2008; Zock 2013). And while religious contexts often help give answers to people about existential questions such as *What is good? Why is there evil? What should I do about it?* (Fivush 2013), religious contexts can also constrain and limit one's openness to spiritual experiences (Sosteric 2018; Zock 2013). Beyond religious experience, what we call spiritual is vast, encompasses a huge swath of human experiences, and often "shatters any box of mental understanding we have been [using]misusing" (DeLay 2021, p. 102). So, I was reluctant to approach this study with an overly well-conceived idea of *spiritual* solely based on my religious upbringing that might incline me to see (or not see) in my experiences what I already know (or think I know) about spirituality.

The words often used to describe *It* (spiritual experience) include epiphany, purpose, meaning, enlightenment, awe, and inspiring. *It* is often described as connecting to something transpersonal, that takes us out of the confines of our ego and brings us back toward one another. Attributes of a spiritual experience might also include something sensory or perceptually extraordinary; a peak or heightened performance; a sense of wholeness, integration, or cosmic harmony; an abundance of heart qualities (compassion, love, kindness, joy); a sense of sacredness and ecstasy; being endowed with clarity, insight into things; and charged with creativity and vitality (Bai et al. 2016, p. 79). Or *it* may be more simply any experience of connection that moves me beyond the "Normal Consciousness of our daily existence" and links us to a mystical sense of *more* as "it leaves one in a *more* harmonious relationship (i.e., connection) to the divine" (Sosteric 2018, p. 21, italics added). *It* is the something *more,* beyond me, as important as me, and definitely including me, but also including everyone and everything else; "*It* is a '*moreness'* that takes us by surprise when we are at the edge and end of our knowing . . . " (Huebner 1999, p. 403, italics added). *It* is a *moreness* that brings an other into clearer focus and "brings new relational possibilities into view" (Poulos 2010, p. 54). *It* opens the possibility for harmonious relational connection sometimes resulting in "spiritual liberation" as "an act of relationship, a sharing in community by one for all" (Denton 2005, p.768). *It* is not a way of believing or doing, but primarily about a way of being compassionately interconnected with all life (Peabody 2022). A vast and varied description, to be sure.

Because spiritual experience is subject to very individual conceptualizations, it makes sense that it is best understood through exploring individual experiences (Tisdell 1999). By looking to individual experiences we can move beyond the "rhetoric of religious dogma" and into "the richness of unscripted human experience" (Denton 2005, p. 755). The 19th century philosopher and theologian William James was strongly committed to studying individual experiences as the most "precise way of understanding mystical, transcendent, spiritual realities" (Hart 2008, p. 517). Reading James (1902) *The Varieties of Religious Experience,* it is easy to conclude that James looked to extraordinary experiences of individuals, including his own, to better understand the mysteries surrounding the transcendent. James practiced what Catholic priest Richard Rohr (2019) called the *sacramental principle,* the

practice of turning to the portals of our everyday lives, hoping that "the visible and tactile" will become "the primary doorway to the invisible" (p. 29). As we look to these portals, we sometimes find ourselves "wrapped up in a moment in which something sacred, something mysterious, something numinous (i.e., the divine) is showing itself to us. This is the kind of experience some of us seek, and some of us just stumble upon" (Poulos 2010, p. 49). The expectation that we might stumble upon what is spiritual and supernatural through our ordinary natural experiences in the physical world positioned me well for the possibility of spiritual encounters everywhere, including in autoethnographic encounters.

## 2. Looking to Autoethnography

I turn now to my journal entries from three virtual autoethnography conferences I attended between January of 2021 and January of 2022 as my main source of autobiographical data. Since completing my autoethnographic dissertation in 2018 I have become accustomed to recording various life experiences that stand out to me by writing in a reflective journal. With the awareness that any autobiographical data can become useful for future autoethnographic explorations, I wrote with detail about what I was seeing and feeling, letting thoughts flow freely without editing. Sometimes I write out meaningful interactions of what was said in a particular workshop word for word, and sometimes in summary form, or in prose, sometimes in rants of confusion, or exclamations of agreement. Writing in real time like this helped me to hold on to my gleanings from experiences so I could revisit them later and add to them after some time of reflection, letting the impressions speak and further capture any insights as they came.

I then used the principles of creative analytic practice (CAP), that is, writing and rewriting as "a method of inquiry, a way of finding out about yourself and your topic" (Richardson and St. Pierre 2005, p. 923). This practice generated deeper thinking around my experiences. CAP writing often produces the "messy texts" that only take shape through many rewritings, to eventually become a cohesive narrative. As I kept my questions about what makes autoethnography a spiritual encounter in the forefront of my thinking, I stayed aware through cycles of critical reflection and writing that my discoveries would not necessarily point to a single objective answer; rather, new understandings would form with each rewriting.

The CAP writing process fits perfectly for a study of a spiritual nature. I have come to see that there really is no way to objectively study what is spiritual, mystical, and transcendent. I only have the experiential proofs in my own experiences. Through a contemplative, open, curious posture I could begin to understand and perhaps even answer for myself what makes autoethnography something extraordinary, mysterious, transcendent, spiritual.

The vignettes[2] below are based on portions from my reflective journals from the virtual autoethnography conferences that have been woven into a coherent narrative to portray a story of my coming to see autoethnography as a practice for spiritual connection. The discussion that follows the narrative portion will help to expand the space for new understandings to be co-constructed between me as the writer and you as the reader, about what autoethnography can be as a spiritual practice.

## 3. Vignettes

International Symposium of Autoethnography and Narrative (ISAN) 2021

*It is January 2021, almost a year into the COVID-19 pandemic. I am getting way too accustomed to my semi-reclusive self. This is distinctly a new me, and I am kind of liking it. The old me was inviting clients into my home office, riding my bike to weekly gatherings at our local Turkish church, and regularly doing life in the company of "others." The two conferences I had been looking forward to attending in person in 2020 were of course canceled. But I settled for virtual participation, recording my presentations, and also sharing in real time over Zoom. It was not too bad. This ISAN 2021 is my third virtual conference since the beginning of Covid. I am starting to be really comfortable*

*attending events that previously seemed off limits to me due to distance, cost, and because I tend to keep going to the same conferences every year. I have been branching out, and this is my first autoethnography conference. Sometimes being an outsider is awkward in person; I wonder what it will be like over Zoom.*

*** 

*It is Day 3 of ISAN 2021*

*This is a very different conference! It's people, over Zoom, sharing their stuff, just like at the other virtual conference I've been to, but it's distinctly not like the other conferences.*

*Today I attended the presentation Betweener Autoethnographies: A path toward social justice with presenters* Diversi and Moreira (2018). *When Marcelo and Claudio (first time "meeting them" but I feel like they would want me to address them as friends) were reading excerpts from their book, Betweener Autoethnography (*Diversi and Moreira 2018*) I was saying Amen out loud, as if the passages were words from friends, reaching me in a far away land. I even gasped a few times I was so touched. They were speaking about what it means to lead a life of "betweener autoethnography" and how a betweener path is a life of vulnerable sharing, truth seeking, speaking out about injustice, describing how it is felt in the body, mind, and spirit, and how it calls to listeners/readers for a response. Another gasp from me, with my hands crossed over my heart.*

*Throughout the session I notice people commenting in the chat boxes that they felt love, saw friendship, were inspired, moved. Those seem like spiritual states to me. So! I am not alone in sensing this was a spiritual encounter. I didn't think so. I'm not sure why but, I sensed what happened there, in that session was something of a spiritual nature.*

*I immediately think of Martin Buber's I-Thou encounter as a bridge to visit the Eternal Thou, that is God, the Sacred Presence. Yes, that is exactly what this was like for me. And now I think others who have written about this type of encounter (Rumi, Jesus, Proust, Friere). Is betweener autoethnography another way of connecting to the Eternal Thou through the sacred act of sharing truth and inviting witnesses?*

*Last Day of ISAN: I needed this. It was the proverbial "breath of fresh air." The Covid air of keeping to myself, staying six feet away, mouth covered, afraid, was replaced. But by what? What was that?*

*I felt like I knew all of those presenting their autoethnographic stories. Even though I was not presenting my story I felt seen inside of theirs, being inside of something familiar, that connected us. It was a moment of true encounter, connection, something that I was used to feeling elsewhere. Maybe in church? Or at a concert where the minor chords and crescendos make me cry? Or during a poetry reading when the rhythm or the rhyme carries me off to someplace that can't be explained?*

*Yep, that's what happened there. I was so moved each day, sometimes crying, sometimes laughing, enjoying the beauty of human connection like I had not encountered before, even at the other virtual academic conferences. Is it just the contrast to the loneliness of Covid?*

*The words "That was spiritual" concluded my entry.*

*It was 10 months before I experienced something similar. The next time it was at the 6th Annual Critical Autoethnography Virtual Conference in October 2021. The theme was This BUBBLE Moment. "Oh, what a cute theme. This should be fun, interesting . . . "*

### Critical Autoethnography October 2021: This BUBBLE Moment

*Today is the first day I was able to attend the conference. It is located in Australia and the time difference threw me off. I missed a whole day of it already and I feel sad. But I entered my first session, surprised by the creativity of the descriptions of the presenter's bubble moment. It is Ellissa Foster, on the fragility of life as she lamented the death of a goldfish entrusted to her care. The realization that "bubbles, like fish are not built to last*

*and no matter how hard we try, the bubbles of safety are a delusion, an illusion of control, that rituals of safety will protect us from murder, conspiracy, pandemic, racism...death. Because we live our destinies tied together. Maybe bubbles last when they're tied together, bump up against each other, providing support, and like this conference we are all bubbles bumping against each other in solidarity." Gasp, my heart is full.*

*Then Dan Harris responds to the story with "it's the emotion that sticks with the everyday that we identify with," such simple words, but deeply felt in me; I realize my head is nodding continuously. Christopher Poulos chimes in, surrounded by the green of his room and a sweet dog laying on his bed, watching him. Chris shares a story about the death of his childhood pet, a gerbil I think. And Zoom! I was transported to my own ungrieved moment of a traumatic childhood memory when I accidentally stepped on my hamster Ollie, killing him (his brother Stanley survived). I can't believe I never really grieved that before. Uhhhh, these bubble moments are not cute. They're heart wrenching, joy giving, hope filled, bubbles of connection, to each other's stories, emotions, hearts, spirits. I guess it's a little bit cute, but mostly it's deeply spiritual.*

<p style="text-align:center">***</p>

*It is only the second presentation of my first day, and Shoot! I'm crying again. I'm listening to Jonathan Wyatt read his autoethnographic prose. He was describing a therapeutic encounter he had over Zoom; saying something about the presence of the therapist, "hearing and seeing her has an impact upon me . . . I feel present, connected, something is rising, looking for meaning." That is exactly what is happening to me as I listen to him. I want to reach through the screen and hug him. Instead I click off my video, because I'm crying. Feeling like a coward I click back in and just look into all the faces of the others on the screen, to see if they are crying too. Some are I think.*

*How did his words touch that place normally reserved for the sacred spaces, like a cathedral, or my pillow at night after prayers. I have the sensation of rising upward and connecting with them from the place where my personal experience meets theirs. It is a sacred space between us: me him, all the other faces listening in and sharing this moment.*

*There is more. Bozz Connelly has magic and cards and the magic of stories is breaking down barriers. We're feeling the sense of "two or more" gathering together through story, and there IT is, the divine sacred presence, in the midst of us.*

### International Symposium of Autoethnography and Narrative (ISAN) January 2022

*It's January 2022 ISAN. I think I know what I'm looking for this time. It's the familiar faces, being greeted knowingly by a few, being invited into intimate and vulnerable moments through stories, there's some reminiscing among the foremothers and forefathers of autoethnography (Denzin, Ellis and Bochner sometimes sitting next to each other, Richardson shows up in a Zoom window, as does Pelias[3]), really all of them who I've been following and knowing through their writing for years. I'm reminded of a book in our family library when I was young. It was called Jewish Heroes of the Faith, and I read it over and over trying to understand my Jewish faith, and who all went before me. I realize I'm bracing myself with readiness to hear from them like they are my heroes of "the faith". Haha I would never tell them that, it feels too much like idol worship. But it seems significant to me.*

*The session was great, a history lesson, of where we've been and where we're going, as autoethnographers. I feel I've been in the presence of superstars. Or maybe it's more like a good old spirit-filled church service. And without fail, I got it. Spiritual Connection. I'm connected, I'm with them, I'm one of them.*

*In each session I follow the Zoom link and hear and say words like "I was so inspired," "your creativity moved me," "what courage I see in your writing." It seems to me there is a feeling of being enlightened, maybe lighter, more movable, empowered to do something good in the world. A corporate response to what this conference meant to us - and*

*everyone said amen, and went in their own directions, to their part of the world, to change it, to make disciples.*

<div align="center">***</div>

*For the next few days, even when I'm talking to Haluk at home or meeting when Caleb and Megan come for dinner, it's autoethnography this, autoethnography that. I'm like a fanatic.*

*I realized later at the dinner table I sound like I've been in some kind of religious big tent meeting and I've had a spiritual revival, a re-awakening. A spiritual connection, made from sipping from the same cup of autoethnographic experiences. That's what communion is right? Connection around the remembrance of who we are, to each other, and in connection with all that is, the Eternal Thou, that is God.*

## 4. Looking to Buber

Martin Buber in his exquisite book of inspired prose called *I and Thou* (Buber [1923] 1996) centers spiritual experience on dialogical encounters between individuals. Buber's central theme is based on his opening words of "In the beginning is the relation." In the sacred space of vulnerable exchange, mutual honor, and acceptance lies the sacred relation. This is the space in which spiritual connection occurs. It is a spiritual space that comes into being if we look for it, and practice seeing and being in it.

Buber distinguished between two fundamental ways of being in the world. As primarily relational beings we are either in an *I-It* or an *I-Thou* relation (Buber [1923] 1996, p. 21). An *I-It* relation is one where the *I* approaches an *It* as an object that is fully separate from the *I*. *It* can be enjoyed, used, discarded, ignored, minimized, forgotten. An *I-It* relation lends itself to seeing an other as less than, as fragmented from, and can lead to labeling, discrimination, and dehumanization.

In contrast, an *I-Thou* relation occurs in the relational space where one's whole being is drawn to honor the *otherness* of an other. In Buber's view, all such *I-Thou* encounters can ultimately turn into connections with the *Eternal Thou*, Buber's conceptualization of God. An *I-Thou* encounter allows us to see an other as a reflection of God. Buber considered that every time someone honors another as a *Thou*, they are indirectly addressing God (Friedman 1976; Mendes-Flohr 2019). They are recognizing the sacred that is in another, creating a spiritual space where the transcendent presence comes into being. That is, God the transcendent presence shows up the moment you engage in an *I-Thou* manner. Such encounters erase the *I-It* propensity to use and oppress an other, and instead makes equals of *I* and *Thou*, summoning the presence of the *Eternal Thou* as what connects us to each other, and enables us to live in our shared humanity.

Buber's *I-Thou* premise helps me to make sense of those unusual moments of human connection in the autoethnographic exchanges described in my narrative. As one's authentic being is seen, recognized, and honored by an other, the experience transcends from ordinary to extraordinary, from natural to supernatural. I venture to say that all of the components for spiritual connection as Buber puts forth can be fully actualized through autoethnographic work.

## 5. Looking with Buber to Autoethnography

*Autoethnography is at its core relational-*

*It is not I-I (that would be solipistic-narcissitic)*

*It is not I-IT (that would be using others are a means to an end, my end, with no reciprocity or movement between us)*

*At its best, autoethnography is an I-Thou practice.*

*An exploration of I that is inter-dependent on Thou (ISAN 2021 Reflective Journal)*

Just as Martin Buber's ([1923] 1996) *I and Thou* opens with the well-known and beautiful words "In the beginning is the relation," the self in relation to others is a core feature of

autoethnography ([Hernandez et al. 2022](#)). The relational, intersubjective *I-Thou* encounter cannot be fully obtained in the *I* alone or in the *I-It*. Something will always be missing. Buber considered that human beings are not isolated, free-floating objects, but in perpetual, multiple, shifting relationships with other people, with their lived experiences, and ultimately with God. Dialogical engagement with the stories of others is what moves us closer to others, and subsequently moves us more intentionally toward spaces where spiritual connection is possible. The autoethnographer who embraces vulnerability and presents "her or himself as an intentionally vulnerable self" can "create a reciprocal relationship with, and compel a response from the reader" ([Jones et al. 2013](#), p. 25).

This is not to say that other forms of scholarship do not seek a response from readers, rather "that autoethnography explicitly acknowledges, calls to and seeks contributions from audiences as part of the ongoing conversation of the work" ([Sparkes 2018](#), p. 483). The invitation into relationality is what separates a solitary reflective piece of writing from an autoethnographic piece. The presence of an *Other* is needed for the resonance that characterizes the *I-Thou* encounter. In the autoethnographic process, others are invited to bear witness to an experience such that connection between *I* and *Thou* is made possible.

Ron [Pelias](#) ([1999](#)) characterized the invitation to bear witness as "a generosity of spirit," wherein the autoethnographer opens a space to "feel with others, to understand what others see" (p. xiii). Peter [McIlveen](#) ([2008](#)) described the same generosity of spirit as empathy, saying that when we invite others into our story, we co-create a "genuine understanding, a shared humanity that reaches across, touches; and in feeling with the other, we become our own self" (p. 19). An empathic connection, wherein the one who invites and the one invited can both imagine and perhaps create different, better lives is crafted.

Autoethnography uniquely stimulates empathic connection in a way that rationality cannot achieve. Outside of autoethnography, researchers do not typically or purposefully create such spaces for emotional empathic encounters. However, encounters of this nature naturally occur through the autoethnographic process as one is investigating the depths of an experience with an open heartedness that lends itself to the sacred and spiritual connections that follow ([Denton 2005](#)).

Autoethnography creates the empathic connection that operates as a bridge between human experiences and sanctifies the space between people. The autoethnographer crosses that bridge through vulnerability and gives voice to the depth and breadth of one's individual experience, then invites another to come along and bear witness as a collaborator, helper, healer. Being so fully present to another is an encounter of bodies, souls, and spirits that lifts us into another dimension. [Lincoln et al.](#) ([2000](#)) were right when they suggested two decades ago that we were "entering an age of greater spirituality within research efforts" (p. 185). Autoethnographers are certainly proving them right.

## 6. Looking to Autoethnography as a Spiritual Research Paradigm

After reading and rereading my reflections from the autoethnography conferences, and holding my impressions all the while, I contemplated *What made these academic gatherings so much like spiritual encounters?* My answer: Connection of the *I* and *Thou* variety, with autoethnographic offerings piling on top of each other to become a bridge straight into sacred spaces. The world needs these offerings, sacrifices of love, vulnerability, truth telling. Autoethnography is an invitation to think with, feel with, be with, and act with. It is a methodology of showing up and showing oneself to an other expecting to be seen, understood, interconnected. Autoethnography's power is felt in the "encounters with Otherness" ([Bochner 2013](#), p. 53).

Encountering otherness is the primary mechanism for movement from *I-It* to *I-Thou* relations. I was drawn to autoethnography before I really understood why. Today, I can say, I get my own attraction to it. As a researcher and someone generally interested and engaged in spiritual inquiry, it makes sense that I would be drawn to methods of inquiry that stretch me beyond bifurcated categories toward a methodology that dissolves the fabricated boundaries of positivist inquiry. In their search for a suitable spiritual

research paradigm (SRP) Lin et al. (2016) lament that "Current research paradigms, due to their limitation to empirical, sensory, psychologically, or culturally constructed realities, fail to provide a framework for exploring this essential area of human experience (p. x, *Introduction*)" However, they may be missing that autoethnography, as a research method that centers on the lived experience of the researcher "as a humanizing, moral, aesthetic, emotion-centered, political, and personal form of representation" (Bochner and Ellis 2016, p. 47), already contains what they are looking for in an SRP.

I have so far suggested that autoethnography is a method of inquiry and a spiritual practice. Autoethnography is no longer relegated to fringes or inquiry for a reason. It can be expanded even more to encompass a wider array of lived experiences that are accessible for exploration by scholars and practitioners. We needed a movement that is welcoming of those whose experiences were previously relegated to the margins, and in autoethnography those experiences can move from the margins to take up the whole page.

## 7. Final Thoughts

Nearly a hundred years ago Martin Buber ([1923] 1996) wrote so beautifully on the sacredness of the relational eternal space that can be found between *I* and *Thou.* Between Me and You, and between Us and Them. But only if we look for, practice seeing it, and living into the possibility of spiritual connectedness.

Perhaps autoethnography was an unusual place to look for spiritual connection. But autoethnography has become many things to many people: a way of life (Bochner 2020), social activism (Bilgen 2018; Holman Jones 2019), an expression of solidarity (Bochner and Ellis 2016), an act of resistance (King 2019; Pławski et al. 2019), truth telling (McIvor 2010), therapeutic (Custer 2022), transformative (Custer 2022; Hernandez et al. 2022), soul work (Callier and Hill 2021). And now I add, it has become a spiritual practice (Rafi 2021).

It is through *I-Thou* connections that inquirers can tap into a dynamic integration of body, soul, and spirit as equally valuable avenues toward a deeper understanding of a full range of human experiences. And as typical of those who have spiritually enlightened encounters, I take the annoying role of evangelist for something I know you ultimately can only experience for yourself. Art Bochner (2013) wrote about autoethnography's *existential calling;* and I believe it is that which keeps calling me. It is the same call I hear in Buber's ([1923] 1996) words, "I become through my relation to the Thou; as I become I, I say Thou. All real living is meeting" (p. 11).

What is it we want from our research into spiritual experiences? Perhaps it is the spiritual experience itself. Sometimes I wonder if all of our spiritual inquiries are not reflections of our human effort to not feel so utterly alone. I know that I want the mutual attunement, empathic wonder, and collaborative synergy to build a better world through my autoethnographic work. I want the *I-Thou* connection that leads into the *Eternal Thou* where I am hopeful that something better awaits.

**Funding:** This research received no external funding.

**Data Availability Statement:** Not applicable.

**Conflicts of Interest:** The author declares no conflict of interest.

## Notes

1    The words spirituality and spiritual experience are at times used interchangeably throughout.
2    Vignettes are italicized throughout and left in an informal vernacular in which they were originally written in the authors reflective journal. There were many other portions of narrative in my reflective journal that I could have included, that speak into the power of work by the many autoethnographers who offered their work at each of the autoethnographic conferences that I write about here. Time and space simply do not allow to name and honor them all!
3    That is Norman Denzin, Carolyn Ellis, Arthur Bochner, Laurel Richardson, Ron Pelias, all who have been adding to the autoethnographic world for years.

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
