# Peer review of "Looking to Autoethnography as Spiritual Practice"

_religions, doi:10.3390/rel13080699_

Round 1
Reviewer 1 Report
I found the essay on “autoethnography” to be conceptually and analytically — if you would allow the hyperbole — brilliant. It presents a daring thesis about the “spiritual,” that is, numinous dimension of intercultural encounters that a grounded in a dialogical attitude as explicated by Martin Buber’s phenomenological meditations on beholding the other as an autonomous “fellow” subject. The author’s lucid, nuanced exposition of her/his autoethnic experiences enhances the compelling depth of the essay. In a word, the essay is worthy of publication without any revisions.
Author Response
Thank you so much for reviewer my work and providing enthusiastic support. I have made proof reading corrections to improve readability even more.
Once again, sincere thanks!
Reviewer 2 Report
This is a compelling and beautifully written paper that draws the reader in from the outset. The author is clearly well-read in the area being discussed (autoethnography) and brings in relevant voices to support the argument being made, combining this with very appropriate journal entries. The passion of the author (for both spirituality and autoethnography - perhaps these are the same thing) comes across strongly in the article.
I wouldn't want the author to make too many changes to the article, as I believe that it reads very well. However, I would suggest that the final substantive section ("Looking with Buber to Autoethnography") could do with some further editing, with perhaps some thought given to what the main points of this section are, and then grouping these together (perhaps under further subtitles?) or at least being clear on what the 3 or 4 main points of the section are. At the moment, it slips a little bit into a (very enthusiastic) stream of consciousness kind of style. It feels like the author has taken their different notes on what autoethnography is and does and how this relates to Buber's I-Thou, and strung these together in this section, rather than structuring the section around a more coherent narrative. Hence, I think it is necessary to clarify what the main points are that the author wants to make in this section, and to develop these clearly. This may mean excluding some other points (which can always be made in future articles). It feels like at the moment the author is trying to put in all their different thoughts into this one article (hence, the feeling of enthusiasm), and stylistically I don't think this works so well.
More clear structure and a stronger central narrative holding it all together would strengthen this final section and take us, as readers, right through to the end.
There are also some spelling and minor grammar issues to address throughout the article.
Author Response
Thank you so much for your thoughtful and very helpful review.
I have edited the section Looking with Buber to Autoethnography to remove the parts that read like a stream of consciousness, dead wood as I now see it. I did not give separate headings in this section however I have edited it way down and I believe this helps the flow an clarifies the main point. In the editing I did exclude some sections and will consider including those in another paper as you have suggested.
I also did some proof reading to correct for grammar, typos and syntax errors.
Thank you again for your really helpful comments that pared this piece down and improved the concise readability!